# Personalized Biomarkers and Neuropsychological Status Can Predict Post-Stroke Fatigue

**DOI:** 10.3390/brainsci13020295

**Published:** 2023-02-09

**Authors:** Hanwen Zhang, Guidong Liu, Li Zhang, Wenshi Wei

**Affiliations:** Huadong Hospital, Fudan University, No. 221, West Yan’an Road, Shanghai 200032, China

**Keywords:** acute ischemic stroke, fatigue, fatigue severity scale, biomarker, anxiety, depression, sleep disorder

## Abstract

Post-stroke fatigue (PSF) is a common complication of stroke that has a negative impact on prognosis and recovery. We aimed to investigate the relationship between PSF and demographics, mood disorders, sleep disorders, and other clinical characteristics of patients with stroke. In this exploratory cross-sectional study, we collected data on sociodemographic characteristics, biological indicators, and imaging features and evaluated patients using neuropsychological scales. Patients were assessed using the Fatigue Severity Scale, Hamilton Depression Rating Scale, Hamilton Anxiety Scale, and Pittsburgh Sleep Quality Index. Magnetic resonance imaging scans were primarily used to evaluate infarctions and white matter lesions. The correlation between the PSF of patients with stroke and clinical indicators was obtained by logistic regression analysis and power analysis. We observed an independent association between fatigue severity and female sex (odds ratio [OR], 2.12; 95% confidence interval [CI], 1.14–3.94), depressive state (OR, 1.50; 95% CI, 1.01–1.73), and sleep disorders (OR, 1.58; 95% CI, 1.01–1.98). High levels of blood glucose, serum uric acid, and homocysteine and low levels of serum triiodothyronine were strongly associated with poor functional outcomes in patients with stroke. Further studies are needed to elucidate how specific structural lesions and anxiety symptoms are related to early PSF.

## 1. Introduction

Stroke is the leading cause of death in Chinese adults [1]. A large amount of attention has been paid to stroke because of its high disability and mortality rates. Stroke survivors often experience irreversible sequelae that considerably impair their functioning and activities of daily living [2]. Stroke has numerous consequences, including those that may not be apparent but have a large influence on the quality of life of patients (e.g., fatigue and neuropsychiatric symptoms) [3]. Post-stroke fatigue (PSF) is a common debilitating symptom that occurs after an ischemic stroke. The worldwide prevalence of PSF was reported to range from 23% to 85% [4]. PSF can persist for a long time and has a negative impact on the physical activities, functional capacities, social activities, and quality of life of stroke survivors [5,6].

Thus, establishment of an effective treatment strategy for PSF is essential for efficient recovery of stroke survivors. However, a clear explanation of the pathophysiology of PSF, which will serve as the basis for establishing a treatment strategy, is still lacking. However, the reasons why patients develop fatigue are largely unknown. In recent years, increasing evidence has demonstrated that PSF is associated with cognitive impairment, affective disorders, disability, and biological factors [7,8]. In the past, stroke treatment strategies mainly focused on physical conditions such as motor or sensory disorders [9]. With the recent improvements in the quality of life of survivors, the importance of managing emotional and mood disorders, which are non-physical disorders, has also been highlighted [10]. Moreover, since the pathophysiology and treatment application for each type of stroke are different, the factors contributing to the development of fatigue symptoms may differ even after the same stroke, and information on these seems to be lacking [11].

In this study, we aimed to investigate the prevalence of fatigue and neuropsychiatric symptoms in patients with acute minor ischemic stroke and to analyze the relationship between neuropsychiatric symptoms (sleep disorders, anxiety, and depression) or endocrine disorders and fatigue in this patient population. A better understanding of the associations among these factors will help in preventing and managing fatigue, which will benefit stroke survivors, their families, and society. We hypothesized that PSF level will be significantly associated with these factors.

## 2. Materials and Methods

### 2.1. Participants

#### 2.1.1. Inclusion Criteria

We enrolled 230 patients with stroke who were treated in the Department of Neurology of Huadong Hospital of Fudan University between January 2020 and November 2020. All participants were informed of the objectives, risks, procedures, and significance of the study and signed an informed consent form before study initiation. This study was conducted in accordance with the Declaration of Helsinki and approved by the Ethics Committee of Huadong Hospital of Fudan University (approval no. HD20191213). The requirement for written informed consent was waived by the Ethics Committee of Huadong Hospital of Fudan University owing to the retrospective nature of the study and the anonymized evaluation of registry data. The inclusion criteria were as follows:Age between 40 and 70 years, with no sex limitation;Minor ischemic stroke (National Institutes of Health Stroke Scale [NIHSS] score < 4) within 7 days of onset;Sufficient cognitive ability.

#### 2.1.2. Exclusion Criteria

The exclusion criteria were as follows:Central nervous system diseases other than stroke;Inability to undergo magnetic resonance imaging examination;Significant aphasia or dysarthria;Dementia, defined as a Mini-Mental State Examination score of <17;Severe heart, lung, kidney, or liver condition or a malignant tumor;Recurrent stroke within 3 months of the initial event [12,13].

### 2.2. Data Collection and Baseline Evaluation

Data on the demographic and clinical characteristics of the patients were collected. Information on age, sex, education level, chronic diseases (e.g., hypertension, diabetes, and coronary heart disease), and lifestyle (e.g., exercise, sleep status, work, diet, and smoking habits) was obtained from the medical records. The time since stroke onset was recorded, and the severity of the initial stroke was assessed using the NIHSS. Blood samples were collected 48 h after the onset of ischemic stroke. The serum levels of free triiodothyronine, free thyroxine (FT4), and thyroid-stimulating hormone (TSH) were measured using an electrochemiluminescence instrument (Abbott i2000 (Abbott Diagnostics, Abbott Park, IL, USA), reagents in kit form). Routine blood biochemical tests were performed using an Olympus 5800 automatic chemical analyzer (Olympus, Tokyo, Japan). We performed the clinical diagnoses according to a uniform diagnostic protocol, with criteria such as LDL cholesterol and HDL LDL cholesterol concentrations, using Dutch Lipid Clinic Network (DLCN). Plasma cholesterol, high-density lipoprotein (HDL) cholesterol, and triglyceride concentrations were determined by standard enzymatic methods. LDL cholesterol was calculated by the Friedewald Equation, but a direct LDL cholesterol assay was used in patients with a plasma triglyceride >4.5 mmol/L.

The participants completed self-report questionnaires. The Fatigue Severity Scale (FSS) was used to evaluate the degree of fatigue. A higher score indicated more serious fatigue and a score of >4 indicated pathological fatigue [14]. Studies that used the Hospital Anxiety and Depression Scale in patients with stroke have indicated that a lower critical score is more suitable for this group [15]. The Hamilton Anxiety Scale was employed to determine whether the patients had post-stroke anxiety. A total score of <7 suggested that the patients did not have anxiety symptoms [12]. The presence of post-stroke depression was judged using the Hamilton Depression Rating Scale. A total score of <7 indicated that the patients had no depressive symptoms. The Pittsburgh Sleep Quality Index was used to evaluate sleep quality, sleep duration, sleep efficiency, and daytime dysfunction within a 1-month period [16]. A score of >5 indicated poor sleep quality. The entire assessment lasted approximately 60 min. Sufficient rest time was allowed between the tests to avoid fatigue-induced bias. The Montreal Cognitive Assessment (MoCA) was used to measure participants’ cognitive abilities. MoCA needs about 10 min to complete and it is widely used in individuals with stroke [17].

### 2.3. Statistical Analysis

Two-tailed tests were used for statistical analysis, and *p* < 0.05 was considered statistically significant. Categorical variables were evaluated using the chi-square test, and continuous variables were analyzed using one-way analysis of variance to assess the associations between fatigue and demographic, medical, and clinical variables. Sample power analysis was conducted to evaluate the reliability of our statistical results (reliability: power of test (1-β) more than 0.8 with significance level = 0.05). The predictive variables in multivariate analysis were based on theoretical importance (e.g., sex) or a *p* value of ≤0.05 in bivariate analysis. Medical and clinical predictive variables that exhibited significant correlations in the univariate analysis were introduced into the multivariate analysis in a stepwise manner. Multivariate analysis was performed using logistic regression, and the effect size of the correlation was reported as odds ratios, or Exp(B), with 95% confidence intervals.

## 3. Results

From the medical records, we identified 300 patients who were hospitalized for treatment in our stroke center between 1 January 2020 and 31 December 2020. As Figure 1 shows, 300 patients were screened, of whom 70 were excluded and 230 were finally included in the analysis. On the basis of the group classification criteria, 73 patients were assigned to the PSF group, and 157 patients were assigned to the no-PSF group. The no-PSF group comprised 110 male and 47 female patients with age ranging from 40 to 85 years (average, 64.30 ± 11.81 years). The PSF group comprised 30 male and 43 female patients with age ranging from 42 to 85 years (average, 66.35 ± 13.37 years).

### 3.1. Comparison of Demographic Characteristics

Table 1 shows the baseline data of patients with minor ischemic stroke stratified into the PSF and no-PSF groups. The sex ratio was significantly different between the two groups (*p* < 0.001). The average age, education level, and proportions of smokers and drinkers did not significantly differ between the two groups. Similarly, the two groups did not show significant differences in the medical histories of hypertension, dyslipidemia, diabetes mellitus, and heart disease.

### 3.2. Comparison of Stroke-Related Characteristics and Mental Disorders

Significant associations were found between PSF and lesion location (thalamus, cerebellum, and brainstem) (*p* = 0.032). Additionally, we found a significant trend toward an increased risk of PSF among patients with anxiety and depression. Stroke survivors with cognitive impairment were more likely to experience PSF than those with normal cognitive function. Detailed findings on stroke-related characteristics, mental disorders, and PSF are presented in Table 2.

### 3.3. Comparison of Biochemical Data between the PSF and No-PSF Groups

The PSF group had significantly higher levels of fasting plasma glucose, homocysteine, and uric acid than the no-PSF group. All blood lipid profile variables did not show a significant difference between the two groups. Endocrinological blood tests revealed that patients with higher FT4 and TSH levels had a slightly higher risk of PSF than those with lower FT4 and TSH levels (Table 2).

### 3.4. Multivariate Stepwise Regression Analysis of Factors Affecting PSF

To identify the independent effects of each factor on PSF, a multiple logistic regression analysis was performed with backward elimination of significant factors in the between-group comparisons. Female sex, sleep disorders, and depressive state were found to have significant negative effects (*p* < 0.05) (Table 2).

## 4. Discussion

Fatigue is a prominent disabling symptom in various medical and neurological diseases, and PSF has been identified as one of the most serious sequelae of stroke in up to 50% of patients. This study involved a retrospective medical record analysis to confirm the characteristics of patients with PSF. Demographic characteristics, stroke-related characteristics, and laboratory test results were collected from patients hospitalized for treatment after stroke for one year at a medical institution. In 2003, De Groot et al. proposed diagnostic standards for PSF [4]. In the present study, the FSS was used to evaluate fatigue in 230 patients with ischemic stroke. The prevalence rate of fatigue in our patients was approximately 31.7%, which is thought to indicate some degree of homogeneity with the participant characteristics in the previous study. The incidence of PSF was reported to vary from 23% to 75%, which can be attributed to the different research methods used and the complex and multidimensional nature of fatigue. Moreover, female patients with stroke were identified to be more susceptible to experiencing clinically significant fatigue than male patients [18]. This is because female patients are more likely to be more eager to return to their normal life and to have a higher demand for a high quality of life. Female patients may pay more attention to discomfort and are more likely to be affected by hormonal changes and stress than male patients [19]. Additionally, younger age, a worse rating of general health at baseline, and low pre-stroke physical activity were significantly associated with higher fatigue levels after stroke.

In our study, higher FSS scores were obtained in patients whose stroke location was the thalamus, cerebellum, and brainstem. However, the stroke location was not significantly correlated with PSF in stroke survivors after excluding confounding factors, which may be due to the small number of recruited patients. In addition, it may be attributed to the complexity and heterogeneity of PSF, which cannot be easily explained by lesion localization. However, researchers have different opinions about the influence of neuroimaging features on PSF. Although many studies have emphasized the important role of stroke location in the pathogenesis of PSF, the existing evidence indicates that infarcts in the cerebral cortex, subcortical white matter, basal ganglia, left cerebellum, right cerebellum, and infratentorial region are closely related to PSF [20].

Mood factors are closely related to fatigue (general or severe) at any time after a stroke. Although fatigue is a characteristic symptom of depression, PSF and post-stroke depression also share common risk factors, such as functional impairments [21]. In addition, anxiety and depression levels remain closely associated with fatigue in patients without depression after a stroke, as also described in studies on other neurological pathologies [22]. Similarly, our study found such an association in participants without a clear diagnosis of post-stroke depression, suggesting that PSF can be affected by symptoms of subclinical depression. Winward et al. reported that one-third of patients develop anxiety or depression within 6 months of stroke onset. Chen et al. demonstrated that baseline depression increases the likelihood of severe PSF in the recovery phase after stroke. Conversely, other reports suggested that PSF and post-stroke depression may be independent symptoms because PSF can occur in patients with stroke without any signs of depression [23]. Moreover, the risk of cognitive impairment is increased by post-stroke depression. As serotonin transporters and brain-derived neurotrophic factors may play an important role in neurogenesis and neuronal survival in brain regions related to learning behavior, memory, and executive function, a decrease in their levels may result in emotional disorders and impaired cognitive function [24]. Furthermore, the relief of depressive symptoms is beneficial to improving the cognitive ability of patients [25].

In line with the findings of previous studies, the bidirectional relationship between depression and insomnia played a prominent role in the occurrence of PSF [26]. However, while the prevalence of insomnia maybe significantly decreased after discharge, the prevalence of PSF did not. PSF can occur without depression and/or insomnia. In this study, we found that acute phase PSF had no significant impact on depression after discharge. However, fatigue in the acute phase that continued to discharge was significantly associated with depression after discharge. Thus, early PSF management can not only prevent PSF, but, also, prevent depression after discharge. Acute phase depression and insomnia had direct associations with acute phase PSF. Additionally, acute phase depression and insomnia also had indirect correlations with PSF after discharge, with the correlation mediated by acute phase PSF. Collectively, these findings highlight the importance of management of both mental and sleep problems during hospitalization in patients with PSF.

In this study, we also tried to determine the relationship between various physical factors and the occurrence of PSF, in addition to psychological factors such as depression. To this end, we analyzed the results of various laboratory tests such as biochemical, endocrine, lipid, general hematological, and inflammatory marker tests. Blood glucose, serum uric acid, and homocysteine levels are closely correlated with the occurrence of PSF [27]. As one of the most important antioxidants in human brain tissues, uric acid has a neuroprotective effect [28]. Transient stress hyperglycemia, a common phenomenon in patients with stroke during the ischemic phase, may aggravate the damage to brain tissues, leading to poor prognosis. Homocysteine may abnormally accumulate in blood vessels under pathological conditions, which can directly or indirectly damage vascular endothelial cell function [29]. This may induce the development of cerebral small-vessel arteriosclerosis, large-vessel atherosclerosis, and thrombosis in combination with other vascular anomalies, thereby resulting in infarction [30]. Our results indicated that patients with PSF were more likely to concurrently have low triiodothyronine syndrome. Other studies have reported that approximately 32–62% of patients with acute cerebrovascular events are affected by low triiodothyronine syndrome [31]. A very low level of serum triiodothyronine leads to a series of negative outcomes, such as endocrine disorders, hormone secretion disorders, lower blood perfusion in brain tissues than the normal physiological level, decreased energy metabolism in nerve cells, and aggravation of nervous system dysfunction, thereby causing fatigue [32]. Some clinical and animal model studies have demonstrated that thyroid hormones may have neuroprotective effects under ischemic conditions. Their neurotrophic effects include reducing the toxicity of glutamate, increasing the density of cerebral vessels, facilitating the transport of glucose to brain cells, and increasing brain connections [33].

This study has some limitations. First, this was a single-center study with a small sample size, and sampling and selection biases may be present, thus, future studies with larger sample sizes are needed. Second, some patients were excluded because of speaking difficulties or unclear consciousness; however, these patients may have experienced a more severe stroke. Third, cognitive assessment scales were used to evaluate the outcomes in our study, but there was a lack of objective measurement tools. Further studies may address these shortcomings in the future by increasing the sample size or by using a longer-duration intervention and follow-up.

## 5. Conclusions

Fatigue is one of the most common depressive symptoms experienced by stroke survivors. The present study demonstrated that fatigue and emotional disorders are common in patients with stroke. When treating patients with stroke, evaluating fatigue, anxiety, depression, sleep quality, and endocrine dysfunction may enable a more targeted therapeutic approach. In clinical practice, better management of PSF is beneficial for improving the quality of life of patients with stroke.

## Figures and Tables

**Figure 1 brainsci-13-00295-f001:**
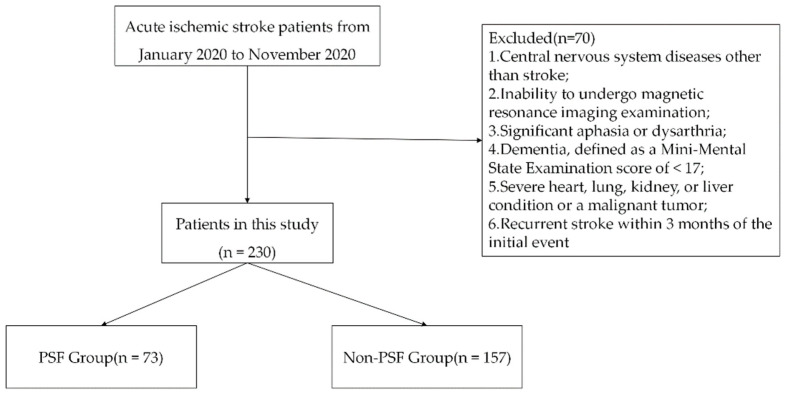
Flow diagram of patient recruitment within the stroke registry data. PSF, post-stroke fatigue.

**Table 1 brainsci-13-00295-t001:** Comparison of characteristics between the PSF and no-PSF groups.

Characteristics	PSF Group	No-PSF Group	*p*
Sex, *n* (%)			<0.001 **
Male	30 (41.1)	110 (70.1)	
Female	43 (58.9)	47 (29.9)	
Age, mean (SD), years	66.35 (13.37)	64.30 (11.81)	0.060
Marital status, *n* (%)			0.479
Single or widowed	15 (20.5)	17 (10.8)	
Living with spouse	58 (79.5)	140 (89.2)	
Education, *n* (%)			0.144
Secondary school or lower	27 (37)	60 (38.2)	
High school or higher	46 (63)	97 (61.8)	
Smoking, *n* (%)	31 (42.5)	80 (51.0)	0.266
Drinking, *n* (%)	24 (32.9)	50 (31.8)	0.315
Medical history, *n* (%)			
Hypertension	57 (78.1)	124 (79.0)	0.065
Diabetes mellitus	18 (24.7)	40 (25.5)	0.114
Hyperlipidemia	10 (13.7)	20 (12.7)	0.265
Heart disease	7 (9.6)	16 (10.2)	0.727

PSF, post-stroke fatigue; SD, standard deviation. ** *p* < 0.001.

**Table 2 brainsci-13-00295-t002:** Comparison and logistic regression analysis of biochemical data between the PSF and no-PSF groups.

Laboratory Parameters				PSF Group	No-PSF Group	*p*
Lesion location, *n* (%)						0.032 *
Cerebral cortex				10 (13.7)	28 (17.8)	
Limbic system				6 (8.2)	16 (10.2)	
Basal ganglia				32 (43.8)	67 (42.7)	
Cerebellum				15 (20.5)	25 (15.9)	
Brainstem				10 (13.7)	21 (13.4)	
Anxiety (HAMA score ≥ 7), *n* (%)				38 (52.1)	36 (22.9)	<0.001 *
Depression (HADA score ≥ 7), *n* (%)				41 (56.2)	42 (26.8)	<0.001 *
Cognitive impairment (MoCA score < 26), *n* (%)				36 (49.3)	59 (37.6)	0.677
Sleep disorders (PSQI), mean (SD)				11.4 (4.3)	7.4 (3.6)	<0.001 *
FPG, mean (SD), mmol/L				8.85 (1.96)	5.86 (1.65)	<0.001 **
TG, mean (SD), mmol/L				1.72 (0.76)	1.66 (0.74)	0.563
TC, mean (SD), mmol/L				5.23 (1.09)	5.02 (1.01)	0.647
LDL-C, mean (SD), mmol/L				3.30 (0.89)	2.96 (0.83)	0.450
HDL-C, mean (SD), mmol/L				1.12 (0.27)	1.11 (0.29)	0.797
UA, mean (SD), µmol/L				285.67 (80.70)	332.11 (82.68)	<0.001 **
FIB, mean (SD), g/L				3.55 (0.81)	3.50 (0.80)	0.374
HCY, mean (SD), μmol/L				24.64 (8.47)	16.32 (6.06)	<0.001 **
FT4, mean (SD), pmol/L				17.22 (4.13)	16.89 (3.79)	0.342
TSH, mean (SD), µU/mL				3.00 (1.56)	2.58 (1.26)	0.278
**Variables**	**β**	**SE3**	**Wald χ^2^**	** *p* **	**OR**	**95% CI**
Female sex	0.897	0.294	4.524	0.030	2.12	1.14–3.94
Sleep disorders	0.265	0.142	6.020	0.010	1.50	1.01–1.73
Depressive state	0.398	0.138	5.686	0.002	1.58	1.01–1.98

HAMA, Hamilton Anxiety Scale; HAMD, Hamilton Depression Rating Scale; MoCA, Montreal Cognitive Assessment; PSQI, Pittsburgh Sleep Quality Index; SD, standard deviation; FPG, fasting plasma glucose; TG, triglyceride; TC, total cholesterol; LDL-C, low-density lipoprotein cholesterol; HDL-C, high-density lipoprotein cholesterol; UA, uric acid; FIB, fibrinogen; HCY, homocysteine; FT4, free thyroxine; TSH, thyroid-stimulating hormone; SD, standard deviation; SE, standard error; OR, odds ratio; CI confidence interval. * *p* < 0.05; ** *p* < 0.001.

## Data Availability

The anonymized data analyzed in this study are available from the corresponding author upon reasonable request.

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
