# Peer review of "Personalized Biomarkers and Neuropsychological Status Can Predict Post-Stroke Fatigue"

_brainsci, 2023, doi:10.3390/brainsci13020295_

Round 1
Reviewer 1 Report
Thank you for your efforts. Just please perform an appropriate copy editing for the manuscript.
Author Response
Thank you for giving us the opportunity to submit a revised draft of the manuscript ‘Personalized Biomarkers and Neuropsychological Assessment Could Predict the Post-stroke Fatigue’ for publication in the Brain Sciences. We appreciate the time and effort that you and the reviewers dedicated to providing feedback on our manuscript and are grateful for the insightful comments on and valuable improvements to our paper. We have incorporated most of the suggestions made by the reviewers.
As suggested by the reviewer, we have carefully edited the entire manuscript and the manuscript has been polished by a professional editor before resubmission.

Reviewer 2 Report
the abstract needs to be rewritten in a more professional manner
Introduction
-please add reference/s to the text "In a recent investigation, it was revealed that stroke ranked first among the various 29 causes leading to death and adult diseases in China." from line 29
-please add reference/s to the text "It has been known that numerous consequences would be induced by stroke, includ- 34 ing some consequences that may not be obvious but had a huge influence on the life qual- 35 ity, for instance, fatigue and neuropsychiatric symptoms."
- references in the text- from line 40 need to be written "2,3" instead of "2-3", the same with "4-5" from line 44 check the guidelines
- the scientific articles need to be written more in a non-personal matter, therefore rewrite at least this phrase "Nevertheless, 40 no effective treatment has been provided yet, partly because we lacked understanding of 41 its mechanisms and related risk factors." - line 40
- in these sentences "In the presented paper, we aimed at studying the prevalence of fatigue and neuro- 46 psychiatric symptoms in patients with acute minor ischemic stroke. Moreover, we intended to analyze the relationship between neuropsychiatric symptoms (sleep disorder, 48 anxiety, depression) or endocrine disorders and fatigue among the population.". PLEASE REWRITE THE SENTENCES, it was a poor choice of words.
Material and methods
-why is the study period chosen so long ago? How would affect the study the choice of patients admitted in 2022? especially talking about the pandemic period because in 2020 the restrictions were putting more weight on the difficulties of stroke patients.
-please rephrase "the patients more than 18 years old with stroke 57 diagnosed by computed tomography (CT) or magnetic resonance imaging (MRI) within 7 58 days of onset, were recruited continuously (NIHSS < 4 indicated that the patient was 59 judged as minor stroke). 60"
-please use:
- inclusion criteria were: …
- exclusion criteria were …
-line 70, would it be better to use "were" instead of "was"?
- try to explain the figure in the caption
ADD A HIGHER QUALITY FIGURE, it cannot be read.
results
-line 119: it is easier to not put numbers one near the other
- a table and a figure with all the results would help much more
discussion
the discussion section should present the impact of your results in light of other studies' results
please upgrade the entire section
Describe the contributions of each author
Please add
Institutional Review Board Statement
Informed Consent Statement
Data Availability Statement
Conflicts of Interest
In the references section
- for each reference, use the citing style of BRAIN SCI
overall MUST- modifications
-please use “space” between words in all the manuscript
-please upgrade the English style of writing to a more scientific way
- please correct the spelling mistakes and style mistakes
-also, the guidelines of BRAIN SCI are very clear and it is easily observed the fact that you have not respected it at all

Author Response
Thank you for giving us the opportunity to submit a revised draft of the manuscript ‘Personalized Biomarkers and Neuropsychological Assessment Could Predict the Post-stroke Fatigue’ for publication in the Brain Sciences. We appreciate the time and effort that you and the reviewers dedicated to providing feedback on our manuscript and are grateful for the insightful comments on and valuable improvements to our paper. We have incorporated most of the suggestions made by the reviewers.
Comment1. Abstract
We appreciate the reviewer’s attention to detail, and we have corrected the text as suggested. The abstract has been rewritten in a more professional manner.
Comment2. Introduction
We apologize for the confusion generated by the previous version of the introduction and sincerely hope that our logic is now easier to follow with this new version. The references were added to the text of the introduction according to the comment (Line 27,33, page 1).
Comment3.Material and methods
3.1. Thank you for pointing this out. In the initial trial protocol, we required a sample size of 700 patients. The COVID-19 pandemic has clearly changed the world in the last three years, the consequences on the conduct of clinical trials are manifold, including, but not limited to:
- Concerns about the safety of patients hence reduction and/or delayed enrolment of patients.
- Extension of the duration of the trial, as the study was either slowed down or interrupted during recruitment
- Overstretched hospitals, resources and systems
- Conversion of physical visits into virtual visits
Due to the impact of the COVID-19 epidemic,300 patients were eventually included in this study.
3.2.We thank the reviewer for pointing this out. We have rephrased inclusion criteria and exclusion criteria according to the comment.
3.3.We have modified the figure and hope that it is could be read clearly.
Comment4.Results
We thank the reviewer for pointing this out. We have put all variables together, including the univariate and multivariate results, including the OR, p-value, 95% CI OR etc. So the reader scan sees how the OR changed/adjusted by other factors towards PSF.
Comment5.Discussion
We thank the reviewer for pointing this out. We have revised.
Comment6.Reference
We have corrected each reference according to the guidelines of BRAIN SCI.
Comment7.We have describe the contributions of each author.
Comment8.We have added Institutional Review Board Statement.
Comment9.We have added Informed Consent Statement.
Comment10.We have added Data Availability Statement.
Comment11.We have added Conflicts of Interest.
Comment12.As suggested by the reviewer, we have carefully edited the entire manuscript and the manuscript has been polished by a professional editor before resubmission. We have corrected the spelling mistakes and style mistakes according to the guidelines of BRAIN SCI.

Reviewer 3 Report
My comments:
1. Introduction.
Good.
2. Materials and Methods:
2.1. Can you include the power analysis?
2.2. Authors wrote, "The levels of ... LDL-C and ...HDL-C) were measured by standard methods. Please provide which method and criteria. Is it the DLCN score?
2.3. In the Statistical analyses section, you are comparing PSF and non-PSF, so which variables used one-way ANOVA for comparison?
2.4. In the statistical analysis section, "Independent logistic regression .... were input step by step," Please clarify, independent logistic regression and chi-square will provide you the same conclusion, so why do you have to repeat this?
3. Results
3.1. Table 4 is the multivariate logistic, so is it include all the variables with p<0.05 in Tables 1, 2, and 3? I suggest putting all variables together, including the univariate and multivariate results, including the OR, p-value, 95% CI OR etc. So the reader scan sees how the OR changed/adjusted by other factors towards PSF.
4. Discussion.
4.1. Is there any limitation of this study?
4.2. Good discussions.
5. Conclusions
Good
Author Response
Thank you for giving us the opportunity to submit a revised draft of the manuscript ‘Personalized Biomarkers and Neuropsychological Assessment Could Predict the Post-stroke Fatigue’ for publication in the Brain Sciences. We appreciate the time and effort that you and the reviewers dedicated to providing feedback on our manuscript and are grateful for the insightful comments on and valuable improvements to our paper. We have incorporated most of the suggestions made by the reviewers.
Comment2.1. Thank you for pointing this out. Based on your advice, we amended the relevant section in the manuscript. All your questions are answered below revised manuscript. In the initial trial protocol, we proposed a dichotomisation of the FSS as primary analysis of outcome, requiring a sample size of 700 patients. However, ordinal analysis of outcome data is becoming increasingly more common in acute stroke trials, as it increases statistical power. For this study, funding is insufficient for inclusion of 700 patients with the overall inclusion rate of 14 patients per week. Power analysis showed that with similar assumptions 450 patients are needed using ordinal regression analysis. Due to the impact of the COVID-19 epidemic,300 patients were eventually included in this study.
Comment2.2. Thank you for your question. Some of the text was ambiguous, and we have modified the text to be more clearly. We did the clinical diagnoses according to a uniform diagnostic protocol, with criteria such as LDL cholesterol and HDL LDL cholesterol concentrations, using Dutch Lipid Clinic Network (DLCN). Plasma cholesterol, high-density lipoprotein (HDL) cholesterol, and triglyceride concentrations were determined by standard enzymatic methods. LDL cholesterol was calculated by the Friedewald Equation, but in patients with plasmatriglyceride >4.5 mmol/L, a direct LDL cholesterol assay was used. This phrase was modified according to the comment (Line 84, page 2).
Comment2.3. Thank you for your question. In these regression models, all effect variables were adjusted for potential imbalances, including the following parameters that were clinically significant or statistically significant in the univariate analysis: age, sex, type of stroke, severity of neurological deficits, accompanying symptoms (e.g., depression, anxiety, sleep disturbance, etc.), autonomic nervous system abnormalities, inflammation, and neuroendocrine dysregulation and baseline NIHSS score.
Comment2.4. Thank you for your question. Some of the text was ambiguous, and we have modified the text to be more clearly. To investigate the differences in characteristics according to the presence or absence of PSF, t-tests (cases with normal distribution) or Mann–Whitney U test (cases that do not show a normal distribution) were performed for continuous variables, and chi-squared tests were performed for categorical variables. In addition, multiple linear regression tests were performed using the backward method to identify the factors affecting PSF. In the regression analysis, the independent variables were defined as FSS scores, and the dependent variables included variables with a significance probability of less than 0.10 in the comparison between the groups.
Comment3.1. Thank you for this suggestion. Table 4 includes all the parameters that were clinically significant or statistically significant in the univariate analysis variables ( p<0.05 ) in Tables. We appreciate the reviewer’s attention to detail, and we have corrected the excels as suggested.
Comment4.1. We think this is an excellent suggestion. The limitations have been added to Discussion section (pg. 6, paragraph 5, line 223-227) .

Round 2
Reviewer 2 Report
Congratulations!